# The Unified Protocol for Transdiagnostic Treatment of Emotional Disorders in Children (UP-C) in Portugal: Feasibility Study Results

**DOI:** 10.3390/ijerph19031782

**Published:** 2022-02-04

**Authors:** Brígida Caiado, Ana Góis, Bárbara Pereira, Maria Cristina Canavarro, Helena Moreira

**Affiliations:** 1Center for Research in Neuropsychology and Cognitive Behavioral Intervention, Faculty of Psychology and Educational Sciences, University of Coimbra, Rua do Colégio Novo, 3030-115 Coimbra, Portugal; anacgois14@gmail.com (A.G.); mccanavarro@fpce.uc.pt (M.C.C.); hmoreira@fpce.uc.pt (H.M.); 2Faculty of Psychology and Educational Sciences, University of Coimbra, Rua do Colégio Novo, 3030-115 Coimbra, Portugal; barbarapereirapsi@outlook.com

**Keywords:** unified protocol, children, transdiagnostic intervention, emotional disorders, feasibility study

## Abstract

The Unified Protocol for Children (UP-C) is a transdiagnostic Cognitive-Behavioral Therapy group intervention for children and caregivers targeting the treatment of children’s emotional disorders (EDs). The present study aims to assess the feasibility and acceptability of the UP-C in the Portuguese population using a single-armed design. The participants were 32 children (6–12 years of age) with an ED (anxiety and/or depressive disorder) as a main diagnosis and their parents. All participants received the UP-C intervention and were assessed at pretreatment, midtreatment, posttreatment, and 3 months posttreatment. Children, parents, the clinicians, and an external observer completed questionnaires to assess the feasibility and acceptability of the UP-C (e.g., satisfaction, motivation, and adherence). Children and parents also completed self-report measures assessing the children’s anxiety and depression and its interference and severity. The results of the present study support the feasibility and acceptability of the UP-C in Portugal; low dropout rates, high adherence rates, and high levels of child and parent satisfaction and motivation were observed. Moreover, significant reductions over time in children’s levels of anxiety and/or depression and of its interference and severity were found and were maintained after 3 months of follow-up. These results are promising and warrant a subsequent randomized controlled trial (RCT).

## 1. Introduction

In the past decade, several studies have consistently documented high rates of concurrent and sequential comorbidity of emotional disorders (EDs) in childhood, e.g., [1,2,3]. ED is a nomenclature that groups anxiety disorders (e.g., separation anxiety disorder, generalized anxiety disorder, and social phobia), anxiety-related disorders (e.g., obsessive-compulsive disorder, and post stress traumatic disorder) and depressive disorders (e.g., major depressive disorder) [4,5]. These conditions have been increasingly grouped due to the evidence of high comorbidity between them, leading to a debate about whether these diagnoses are, in fact, distinct disorders or if there is an overlap between them (i.e., if they share the same phenomenological features) [6,7]. This debate has implications not only for the understanding and study of EDs (more focused on transdiagnostic mechanisms) but also for the selection of treatment approaches (transdiagnostic versus disorder-specific).

Currently, there are various disorder-specific evidence-based treatments (EBTs) based on cognitive behavioral therapy (CBT) for youth anxiety disorders and depression, e.g., [8,9]. These EBTs have been strongly supported for their efficacy in treating specific diagnoses, e.g., [10,11], and remain the type of treatment most often implemented by mental health professionals. However, some limitations have been highlighted in these interventions, particularly regarding their (in)adequacy in treating patients with comorbid diagnoses. Specifically, the comorbidity between EDs has been identified as a barrier to the success of traditional EBTs for EDs, being associated, in some studies, with a weaker treatment response [12,13,14]. In addition, as traditional disorder-specific interventions only target one disorder at a time, additional treatment is often required to address common comorbidities, which would imply higher costs for families and health care institutions and greater treatment length [15,16]. Moreover, sequential comorbidity is also frequent, i.e., anxiety disorders (typically occurring in early or mid-childhood) often precede the onset of depressive disorders (typically occurring in adolescence) [3,17]. Thus, another limitation of disorder-specific interventions is that although the literature posits that early anxiety is a risk factor for the development of future depression or other EDs, disorder-specific interventions targeting anxiety disorders do not directly prevent depression or other EDs. In fact, since disorder-specific CBT protocols only address a singular emotion domain (e.g., fear/anxiety), they would have minimal effect on the regulation of other emotions (such as sadness or anger), e.g., [18], which means that children may improve in the regulation of one emotion but still suffer from dysregulation of other emotions.

Another important limitation of EBTs based on CBT for youth anxiety and depression is that some of them are technique-specific, e.g., focused on psychoeducation techniques such as Psychoeducational School Based Group Intervention for Socially Anxious Children [19]; mindfulness strategies, e.g., BREATHE [20], or self-compassion strategies, e.g., Making Friends with Yourself: a mindful self-compassion program for teens [21], which may not be adequate for all patients (e.g., some patients may not respond well to a specific technique or can benefit more from using other complementary techniques), and therapists (e.g., they may need training in different protocols to master different techniques).

Additionally, the traditional EBT for youth anxiety and depression is usually focused on children and has low parental involvement, e.g., Coping Cat [22]; Primary and Secondary Control Enhancement for depression [23]; FRIENDS [24], which is an important limitation since parents are important vulnerability and maintenance factors of children’s EDs, e.g., [25,26].

As a response to these limitations, there has been growing interest in the concept of a common element approach that focuses on identifying specific techniques or components that overlap across multiple EBTs for a specific problem area, e.g., Common Elements Approach [27] and of transdiagnostic approaches, e.g., [28,29].

Transdiagnostic CBT for EDs, in contrast to disorder-specific CBT, targets the simultaneous treatment of more than one ED and the promotion of regulation across multiple emotion domains, addressing the common vulnerabilities and maintenance factors underlying different EDs [4,29,30]. Therefore, transdiagnostic approaches offer clinical and practical benefits compared with traditional disorder-specific CBT protocols. First, these approaches will be potentially more efficacious in the treatment of concurrent comorbid disorders and in the prevention of other future EDs since the intervention targets shared mechanisms of several disorders. Second, they also target the treatment of symptoms that do not meet diagnostic criteria for specific DSM-5 diagnostic categories or subclinical conditions, while a disorder-specific CBT only targets one specific categorical diagnosis. Third, with a single protocol, therapists may be able to treat multiple EDs and to be trained in different techniques. Thus, it is more likely that therapists will be interested in this type of approach, which therefore may increase UP-C dissemination and implementation. Moreover, when delivered in a group format, transdiagnostic approaches allow the inclusion of patients with different EDs in the same group, which increases the heterogeneity of the group in terms of clinical conditions and is more cost effective. Fourth, transdiagnostic CBT also presents practical and economic benefits, especially for patients with comorbid disorders, because it diminishes the treatment length (since one intervention can target different emotional difficulties) and prevents future accessing of mental health care services [15,28,31,32,33,34].

### 1.1. Unified Protocol for Children (UP-C)

The Unified Protocol for Children (UP-C) [35] is a transdiagnostic and emotion-focused CBT treatment for children with EDs (7–13 years of age) and their caregivers, and it is an extension of the Unified Protocol (UP) for adults [4], as well as the Unified Protocol for Adolescents (UP-A) [35]. The UP-C addresses the limitations of traditional CBT treatments by being transdiagnostic (instead of disorder-specific), unifying a set of techniques (instead of being technique-specific), and having strong caregiver involvement in the treatment. Specifically, the UP-C is composed of 15 weekly group sessions of 90 min for children and at least one caregiver. A set of techniques (i.e., psychoeducation, cognitive restructuring, mindfulness, problem solving training, behavioral activation, and a strong component of emotion exposure-based treatment) are applied across multiple emotion domains (e.g., fear, anger, sadness) to reduce children’s anxiety and/or depression symptoms and promote emotion regulation. All skills are learned in an interactive and child-friendly format, using the analogy of an “emotion detective,” relying on experiential and engaging activities [35].

Although limited and only available for the USA population, the results regarding UP-C feasibility and efficacy are promising [36,37,38]. Specifically, in an initial open trial with children who had an anxiety disorder as the main diagnosis (with and without comorbid depressive symptoms), medium to large improvements from pre- to posttreatment were found in the severity of the main anxiety disorder and of depressive symptoms, and 78% of children posttreatment did not meet the criteria for any anxiety disorder. In addition, the UP-C proved to be feasible, e.g., high rates of attendance (i.e., ranging from 3 to 15 sessions; M = 11 sessions) were found; 73% of children reached treatment completer status; and a high degree of children’s and parents’ satisfaction was found at the end of the intervention [36].

More recently, an RCT compared the efficacy of UP-C with the efficacy of an anxiety-specific group treatment protocol, Cool Kids (CK [39]) [37]. The results of this study indicated that UP-C was at least as efficacious as CK, since both protocols evidenced significant reductions in anxiety symptoms from pre- to posttreatment (according to child and parent reports). However, UP-C participants, compared with CK participants, experienced lower levels of parent-rated child depressive symptoms from pre- to posttreatment and experienced faster changes in parent-rated sadness (dys)regulation from pre- to posttreatment and from posttreatment to follow-up. Moreover, cognitive reappraisal did not display any change from pre- to posttreatment for CK participants but displayed a faster increase for UP-C participants from pre- to posttreatment and to follow-up. UP-C also produced greater improvements than CK regarding the treatment response at the follow-up.

### 1.2. The Present Study

Research on the UP, UP-A, and UP-C has shown that the Unified Protocol is an efficacious treatment for EDs in children, e.g., [37], adolescents, e.g., [40], and adults, e.g., [41], presenting several advantages over traditional CBT treatments. However, research on the feasibility, acceptability and efficacy of the UP-C is still in its infancy, and the existing studies were only conducted in the USA. Therefore, additional studies are needed to attest to the feasibility and efficacy of the UP-C in other cultures (such as in Europe).

Given the limited research on the UP-C, particularly outside of the USA, and considering that transdiagnostic interventions for children are not yet well-disseminated in Portugal, the present study aims to examine the UP-C’s acceptability and feasibility among the Portuguese population. This feasibility study examines whether the UP-C is an acceptable intervention for Portuguese children and their parents and determines whether the recruitment and data collection procedures require further refinement before proceeding to an RCT study to examine the efficacy of the UP-C in the Portuguese population. Specifically, the present study aims to assess (a) the feasibility of the recruitment procedures (e.g., participant flow); (b) the feasibility of the data collection procedures (e.g., administration time of the assessment protocol); and (c) the feasibility of the UP-C intervention and study procedures (e.g., resources needed; therapist adherence; skills learned by children). The acceptability of the UP-C by children and parents (i.e., dropout and adherence rates, satisfaction, motivation, engagement in activities, etc.) and the preliminary effects on treatment outcomes (e.g., reduction in children’s anxiety and depression symptomatology levels and the ED’s severity and interference in their lives and their families’ lives) is also examined.

## 2. Materials and Methods

### 2.1. Ethical Issues

Prior to the investigation, approval from the Ethics Committee of the Faculty of Psychology and Education Sciences of the University of Coimbra (FPCEUC) was obtained. Additionally, authorization for sample collection was obtained from the Ethics Committee of the Tondela-Viseu Hospital Center (CHTV; a central hospital in central Portugal) and from the Board of Directors of three school groups of central Portugal that agreed to collaborate in this research project. All parents who participated in the study provided informed consent about their own and their children’s participation. Children verbally assented to participate in the study.

### 2.2. Trial Design

The present study is a single-armed feasibility study. All children who met the inclusion criteria for the study and their caregivers received the UP-C intervention (15 weekly sessions) and were assessed four times: (1) pretreatment (before the beginning of the UP-C intervention, baseline); (2) midtreatment (eight weeks after starting the UP-C intervention); (3) posttreatment (16 weeks after starting the UP-C intervention); and (4) follow-up (3 months after the end of the UP-C intervention).

### 2.3. Recruitment and Procedure

The primary inclusion criteria were being a child 6–13 years of age with a principal diagnosis of an ED (i.e., anxiety disorder and/or related disorder and/or depressive disorder). Exclusion criteria included: (1) children with a diagnosis of schizophrenia or bipolar disorder; (2) children or parents with a diagnosis of cognitive disability and/or pervasive developmental disorder; (3) children’s severe current suicidal ideation; (4) children receiving psychopharmacological intervention that was not stable in the previous month; (5) children receiving other psychological treatment during the UP-C; (6) inability of at least one caregiver to attend the assessment and treatment sessions; and (7) children’s or caregivers’ inability to speak and understand Portuguese. Children with a principal diagnosis of an ED and comorbidities with nonemotional disorders (e.g., attention-deficit/hyperactivity disorder, oppositional defiant disorder, tic disorders, and autism spectrum disorder) were not excluded to increase the sample representativeness and the generalizability of the results to real clinical contexts.

Participants were recruited for the present research project from June 2020 until April 2021 at a central hospital in central Portugal, at three public schools in central Portugal, and through parental self-registration on the project website. Families were referred to the study by mental health professionals of the collaborating hospital and by school psychologists from the collaborating schools, who preliminarily assessed children’s emotional difficulties and parents’ willingness to participate in the study. Parents could also apply to participate in the study on a website that was developed by the research team for dissemination purposes and that was disseminated in social networks and via email. After professionals’ referral or parents’ registration, the research team contacted all parents to provide information about the project and to schedule an initial interview to assess children’s and parents’ eligibility to participate in the study.

In the eligibility interview with parents and children, a clinical psychologist from the research team assessed the inclusion and exclusion criteria through a nonstructured interview (e.g., assessing the reasons for enrolling in this project and the child’s main difficulties) as well as through the Mini International Neuropsychiatric Interview for Children and Adolescents MINI-KID [42,43]. The MINI-KID is a structured clinical diagnostic interview for children aged between 6 and 17 years of age designed to assess the presence of mental illness over the past 6 months, according to the Diagnostic and Statistical Manual of Mental Disorders (DSM)-5 and the International Statistical Classification of Diseases and Related Health Problems (ICD)-10 psychiatric disorders [42,44]. The MINI-KID takes between 30 and 90 min to be administered, and it is organized in diagnostic sections assessing anxiety and related disorders, mood disorders (including suicide risk), tic disorders, disruptive disorders and attention-deficit hyperactivity disorder, eating disorders, adjustment disorders, substance-related disorders, and psychotic disorders, and there is also a section to screen autism spectrum disorder. Since our sample was composed of children aged 6 to 12 years of age, the section assessing substance-related disorders was not applied. Each MINI-KID section started with screening questions for each specific disorder, and additional questions were only asked if the screening questions were positively answered. All questions are dichotomous (“yes” or “no”). Diagnostic criteria are summarized on a summary sheet, allowing the interviewer to decide which disorder should be the main diagnosis taking into account the intensity, impairment, and duration of the symptoms [42]. MINI-KID has demonstrated strong validity and reliability in population and clinical samples of children and youth [42,44,45].

At the end of the eligibility interview, parents were informed whether the child met the eligibility criteria and could be part of the study. Children who met the eligibility criteria completed the baseline assessment questionnaires after the interview.

The UP-C intervention started between less than one week and a maximum of four weeks after the baseline assessment. Six group interventions were conducted (two at the hospital, three at the schools, and one at the Faculty of Psychology and Education Sciences of the University of Coimbra) between September 2020 and July 2021. Most (84.4%) of the children were accompanied at the UP-C intervention only by their mothers, 6.3% were accompanied only by their fathers, and 6.3% were accompanied by both parents.

Children and parents were assessed at baseline (pretreatment), midtreatment, posttreatment, and at the 3-month follow-up using the same battery of self-report instruments. The parent who completed the questionnaires was the one who attended most sessions. Questionnaires were read aloud to children younger than 10 years of age to ensure comprehension and avoid careless responses. Older children (10–13 years) preferred to complete the questionnaires on their own. Additionally, regarding the self-response measures, a clinical assessment of the severity of the child’s symptomatology, the child’s improvement with treatment, and the acceptability of the UP-C at posttreatment was applied. Children and parents were also assessed at each UP-C session regarding issues such as motivation and satisfaction (see Treatment Outcomes section).

### 2.4. Participants

The participants of the present study were 32 children (56.3% boys and 43.8% girls) 6–12 years of age (M = 8.25, SD = 1.83) and their parents (90.6% mothers) 31–49 years of age (M = 40.78, SD = 3.62). As presented in Table 1, the majority of children presented an anxiety disorder or an anxiety-related disorder as the main diagnosis, with only 3.1% of children presenting depressive symptomatology as their main difficulty. Seventy-five percent of children presented at least one comorbid diagnosis. The majority of children presented comorbidity with another ED (59.4%), 18.8% presented comorbidity with Attention-Deficit/Hyperactivity Disorder (ADHD), and 3.1% presented comorbidity with another disorder (e.g., autism spectrum disorder). Specific comorbidities with other EDs are presented in Table 2. Almost all children (90.6%) were not medicated, and those who were medicated had been on a stable dose for at least 3 months when the intervention started (all of them were medicated for behavioral problems or for ADHD).

Most parents were married (59.4%) or were living together (25%). The majority had graduated (43.8% had a university degree, and 6.3% had post-graduate degrees), were employed (90.6%), and had a monthly income between EUR 800 and 2000 (56.3%). Households were composed of 2–6 people (M = 3.87, SD = 0.99), with 40.6% of children having at least one sibling. Regarding parental psychopathology, 43.8% of parents reported having a diagnosis of psychopathology, and 15.6% reported currently receiving psychological or psychiatric treatment. Families were from three districts of central Portugal (Viseu (59.4%); Coimbra (28.1%); Aveiro (9.4%)); 53.1% lived in urban contexts and 40.6% lived in rural contexts.

### 2.5. Treatment: The UP-C

All participants received the UP-C intervention, which is composed of 15 weekly sessions of 90 min for children and for at least one of their caregivers. Sessions were delivered in a group format (4–7 children and caregivers per group).

A set of CBT techniques were applied across multiple emotion domains through the analogy of an “emotion detective” and were organized into five primary sections around the acronym CLUES (“Consider How I Feel,” “Look at My Thoughts,” “Use Detective Thinking,” “Experience My Fears and Feelings,” and “Stay Healthy and Happy,” see Table 3).

Parents participated in all sessions. In addition to learning the same skills as the children did (to generalize the skills to the naturalistic context of the child and learn to be co-therapists), all parental sessions included parental training through which parents learn about maladaptive behaviors in which they might engage and thus maintain children’s difficulties (emotional parenting behaviors), as well as alternative and more adaptative behaviors (opposite parenting behaviors, see Table 3). In the 12th, 13th, and 14th sessions, parents were only enrolled in the beginning and at the end of the intervention since these sessions were focused on exposure techniques with the children.

The majority of sessions began with parents and children together along with the two clinicians for a quick briefing and to review homework assignments (approximately 15–20 min). Then, the clinician assigned to the parents’ group adjourned to a separate room, while the other clinicians remained with the group of children (approximately 60 min). During this time, children and parents learned the specific session skills. Finally, parents and children rejoined to review skills and assign homework (approximately 15 min) [35].

The UP-C sessions were conducted by three clinicians (two doctoral students in clinical psychology with clinical experience in cognitive-behavioral interventions and one intern of a master’s degree in Clinical Psychology (cognitive-behavioral approach)) under the close supervision of a licensed senior team member. Prior to the investigation, the study team received training in the UP-C directly from the original author of the program, and during its administration, the therapists received supervision.

### 2.6. Outcomes

#### 2.6.1. The Feasibility of the Recruitment Procedures

The feasibility of the recruitment procedures was assessed by analyzing the participant flow and by the constraints that the investigation team faced during the recruitment process.

#### 2.6.2. The Feasibility of the Data Collection Procedures

The feasibility of the assessment protocol was assessed through the participant’s qualitative and oral feedback about the protocol and the clinical perception of the psychologists who applied the protocol in terms of total completion time and observed levels of fatigue and comprehension.

#### 2.6.3. The Feasibility of the Implementation of the UP-C and Study Procedures

The feasibility of the implementation of the UP-C and study procedures was assessed by the analysis of the treatment integrity and of the resources that were needed for the implementation of the UP-C, as well as the feasibility and acceptability of the translated intervention protocol. Clinicians also completed questionnaires assessing their opinions on the skills learned by the children undergoing treatment.

Resources needed: The resources needed for the implementation of the UP-C, in terms of the number of psychologists needed for the application of the program and group management, as well as other necessary materials (e.g., office supplies, teaching materials, construction of materials), were qualitatively evaluated by the therapists who implemented the group interventions. The therapists also qualitatively assessed whether it was feasible to apply this protocol in the context of health services and public schools (for example, due to the acceptability and/or constraints of the implementation of the UP-C in these settings).

Treatment integrity: To assess therapists’ adherence to the intervention, after each UP-C session, each therapist completed a checklist with the content/skills they should have addressed in that specific session. The same checklist was completed by an external observer. All skills were rated dichotomously (0 = skill not covered; 1 = skill covered).

Learned skills: To assess the degree to which children learned the skills taught at the UP-C, the research team developed a questionnaire to be completed by the therapists at the end of treatment. This questionnaire included five items assessing the degree to which the therapists considered the UP-C skills: (1) psychoeducation of emotions; (2) problem solving; (3) cognitive skills (thinking traps, flexible thinking); (4) experiential and mindfulness strategies; and (5) emotional exposure techniques were learned by each child; ratings were based on a 5-point Likert scale (from 1 = very good to 5 = nothing/did not learn at all).

#### 2.6.4. Acceptability of the UP-C by Children and Parents

The acceptability of the treatment was assessed through the examination of dropout and adherence rates. In addition, children’s and parents’ commitment to the UP-C as well as their motivation and satisfaction with each session and with the global UP-C intervention were also assessed.

Session-to-session commitment: Two questions assessed whether children and parents completed their homework and whether they talked with each other about the UP-C during the previous week. These questions were answered in a dichotomous format (yes/no) by parents and children at the beginning of every session.

Session-to-session motivation and satisfaction: Two questionnaires were developed to assess both parents’ and children’s motivation for the treatment and their satisfaction with each specific session. These questionnaires were completed after every session. The children’s questionnaire was composed of three questions rated on a 5-point Likert scale ranging from “nothing” to “extremely/everything”, assessing (1) how fun the child found the session; (2) to what extent did the child understand the content covered in the session; and (3) to what extent did the child consider the content learned in the session useful. The parents’ questionnaire was composed of five questions rated on a 5-point Likert scale ranging from “nothing” to “extremely”, assessing (1) how helpful did they find the session content for their children; (2) how helpful did they find the session content for themselves; (3) their current level of involvement/motivation in the UP-C program; (4) their children’s current level of involvement/motivation in the UP-C program; and (5) their opinion on the psychologists’ level of knowledge about the session content and on the psychologists’ adequacy (e.g., technical skills, and social skills).

End-of-intervention satisfaction: An End of Program questionnaire was developed to assess both parents’ and children’s satisfaction with the UP-C program at posttreatment. Each questionnaire included 12 items, answered on a 5-point Likert scale, assessing the degree to which the participant (parent or child) believes the child and/or parent benefited from the program and how well the program was implemented (responses ranged from 1 = totally false/totally disagree, to 5 = totally true/totally agree). Examples of children’s items were “psychologists explained well the things we learned in this program!”, “since I started this program, I think my difficulties have improved”, “I enjoyed participating in this program”, “I enjoyed that my mom/dad participated in this program”, and “I would recommend this program to a friend!”. Examples of parents’ items were “the way in which the psychologists conveyed the program’s contents was adequate and effective”, “I feel the program has given me useful tools as a parent to help my child”, “my child improved his/her difficulties with this program”, and “I would recommend this program to other people”. This questionnaire also included two open-ended questions: “what did you like the most?” and “what did you like the least?”.

#### 2.6.5. Effects on Treatment Outcomes

Symptom severity according to top problems (child’s and parent’s reports): As part of the UP-C intervention, a tracking form was provided at the beginning of each session to both children and parents (Appendix 11.1 of the therapist’s UP-C manual [35]) to define the three main difficulties that led them to seek treatment (or, in other words, the therapeutic goals) and to classify, over the UP-C program’s 15 weeks, the severity of the three main problems that led the child to treatment (on a scale ranging from 0 to 8).

Symptom severity (clinician report): The Clinical Global Impression (CGI) scale [46] provides a brief, stand-alone assessment of the clinician’s view of the patient’s global functioning prior to and after an intervention. The CGI comprises two single-item measures, one of which is the Severity scale (CGI-S), which was used to assess the therapists’ views of the overall severity of children’s problems relative to their past experience with patients who have the same diagnoses. Answers were rated on a 7-point Likert scale, ranging from 1 (normal, not at all ill) to 7 (among the most extremely ill of patients). Higher scores on the CGI-S are indicative of a higher severity of illness. According to previous child anxiety treatment studies, CGI-S scores of “1” or “2” were considered to be indicative of diagnostic remission [16,47].

Treatment improvement (clinician report): The second measure that is included in CGI [46] is the Improvement Scale (CGI-I), which was used posttreatment to assess the therapists’ view of whether the child has improved or worsened relative to a baseline state at the beginning of the intervention. This single-item measure was rated by the clinician on a 7-point Likert scale ranging from 1 (very much improved) to 7 (very much worse). Higher scores on the CGI-I are indicative of poorer improvement with the treatment. According to previous studies, CGI-I scores of 1” or “2” were considered to be indicative of treatment response [10,37].

Child’s anxiety and depression symptoms (child’s report): The Revised Child Anxiety and Depression Scale (RCADS) [48] was used to assess children’s anxiety and depression symptoms. The RCADS has 47 items, rated on a 4-point scale ranging from 0 (never) to 3 (always), and yields two subscale scores: (1) Depression (10 items; e.g., “I feel sad or empty”) and (2) Anxiety (37 items distributed across five domains: separation anxiety disorder (e.g., “I fear being away from my parents”), generalized anxiety disorder (e.g., “I worry that something bad will happen to me”), panic disorder (e.g., “I suddenly become dizzy or faint when there is no reason for this”), social phobia (e.g., “I worry what other people think of me”), obsessive-compulsive disorder (e.g., “I have to do some things over and over again, like washing my hands, cleaning or putting things in a certain order”)). Higher scores in the total scores for anxiety and depression factors indicate higher levels of anxiety and depression, respectively. The reliability of the total score of the RCADS in the current sample was 0.93 at baseline, 0.93 at midtreatment, 0.95 at posttreatment and 0.94 at follow-up. The Cronbach alpha values for the total score of anxiety subscale ranged between 0.091 and 0.93, and those for the total score of the depression subscale ranged between 0.76 and 0.88.

Interference of the symptomatology in children’s and family’s life (child’s and parent’s reports): The Child Anxiety Life Interference Scale for children (CALIS-C) and for parents (CALIS-P) are two parallel measures [49,50] that assess the life interference and life impairment of the main symptomatology experienced by the child, from the child’s (CALIS-C) and parent’s (CALIS-P) perspectives. The CALIS-C (9 items) assesses the extent to which the child’s main symptomatology impacts the child’s school, social, and family functioning (e.g., “how much do fears and worries make it difficult for you to do the following things?”: “getting your homework done”; “being with friends”; “getting on with parents”). CALIS-P (16 items) is composed of two subscales: (1) Child Interference Subscale (“how much do fears and worries interfere with your child’s everyday life in the following areas”), and (2) Family Interference Subscale (“how much do your child’s fears and worries interfere with your everyday life in the following areas”), providing an additional set of items evaluating the interference that the child’s anxiety has on the parents’ life. All items are rated on a 5-point scale ranging from 0 (not at all) to 4 (a great deal). Higher scores are indicative of greater interference of the child’s symptoms in the life of the child and the family. The Cronbach’s alpha values for the CALIS-C were 0.74 at baseline, 0.78 at midtreatment, 0.83 at posttreatment and 0.75 at follow-up. The Cronbach’s alpha values for the CALIS-P were 0.89 at baseline, 0.88 at midtreatment, 0.89 at posttreatment and 0.92 at follow-up.

### 2.7. Statistical Methods

The univariate normality was analyzed; estimates of skewness and kurtosis were within normal limits, as defined by Kline [51]. Descriptive statistics were used to describe the feasibility and acceptability of the intervention as well as the CGI, RCADS, CALIS-C, and CALIS-P mean scores. To assess the effect of time, comparing the pretreatment, midtreatment, posttreatment, and follow-up scores on the RCADS, CALIS-C, and CALIS-P variables, a repeated-measures analysis of variance (ANOVA) was used. Effect sizes for ANOVAs were reported as partial eta-squared (ηp2), for which values of 0.01, 0.06, and 0.14 are considered to reflect small, medium, and large effects, respectively [52]. Additionally, Wilcoxon tests were used to compare the means of CGI-S scores pre- and posttreatment.

## 3. Results

### 3.1. The Feasibility of the Recruitment Procedures

Before the beginning of the present study, the research team planned to recruit participants through collaboration with a public hospital and some private clinics. However, this proved to be insufficient by itself especially given the limitations imposed by the COVID-19 pandemic (for example, during the pandemic period, the hospital did not accept entry of people who did not work there). Thus, some changes were made to the initially planned recruitment procedures. For instance, we decided to invite some public schools to collaborate in the study and to allow self-referral for the project by developing a website project so that parents who were interested in the intervention could enroll their children in the study.

Throughout the recruitment process, the research team realized that a high number of children referred by school psychologists or by parents who registered on the project website were not eligible for this study (see the Participant Flow section and Figure 1). Thus, the research team decided to start making a screening phone call to briefly present the research project, the eligibility criteria, and the UP-C intervention aims to the families. This procedure allowed for an initial screening of cases and reduced the number of eligibility assessment interviews. Additionally, a high number of people who self-registered on the project website did not respond to subsequent emails (see Participant Flow section and Figure 1); thus, the research team decided to request a telephone contact as a mandatory field in the website registration form. This procedure allowed us to not lose so many participants in the screening phase.

With regard to the inclusion criteria, we initially decided to include children between 7 and 13 years of age [35]. However, the research team received many requests to include 6-year-old children with emotional difficulties in the study. Since the UP-C therapist guide indicates that children slightly above or below this age range can benefit from the UP-C, we decided to include children as young as 6 years of age. Therefore, the primary inclusion criteria were reformulated to include children from 6 to 13 years of age.

#### Participant Flow

One hundred and fifteen children were referred to the research project between August 20 and December 20 (9.6% by the mental health professionals of the hospital; 33.9% by school psychologists; and 56.5% by parents’ self-registration on the project website). However, only 52 children (45.2%) were assessed for eligibility criteria. Exclusion reasons are summarized in the Participant Flow diagram (Figure 1). The majority of excluded children were those whose parents self-registered on the project website or who were referred by school psychologists.

As presented in Figure 1, one of the reasons for exclusion at the screening phase was not responding to subsequent emails in the cases of self-registration on the project website (44.4%). Other reasons for exclusion were, in the screening phone call, not reporting any emotional difficulty of the child at the moment (9.5%), having goals incompatible with the goals of the intervention (e.g., treatment for learning or behavioral difficulties or for family issues (14.3%)), or not being able to attend the sessions with the required regularity due to accessibility difficulties (e.g., transportation, schedule incompatibility (20.6%)). Other reasons included refusal to participate because of not wanting to use a facemask during the group intervention (1.6%), and not having interest in participating in this study (9.5%).

Therefore, 52 children (50% referred by school psychologists; 34.6% enrolled through the project website; and 15.4% referred by the mental health professionals of the hospital) were assessed for their eligibility by the research team in a clinical interview that included the structured MINI-KID clinical interview (see Materials and Method section). Of the 52 assessed children, only 34 (65.4%) met the inclusion/exclusion criteria (see Participant Flow Diagram in Figure 1). Children were excluded because they did not have a principal diagnosis of an anxiety, anxiety-related, or depressive disorder (i.e., they had a principal diagnosis of ADHD or oppositional defiant disorder (61.1%), learning difficulties (5.6%), or autism spectrum disorder and comorbid ADHD (5.6%)) or they did not have clinically significant symptoms (e.g., they had normative and noninterfering fears or were facing family difficulties such as their parents’ divorce; 22.2%). One child (5.6%) was excluded for not assenting to participate in the study and denying any emotional difficulties.

Therefore, a total of 34 children met the inclusion criteria for the study. Of these, two children (5.9%, one self-referred and one school-referred) withdrew from the intervention before it started due to unavailability to be present at the day and time scheduled for the intervention. Consequently, 32 children and their parents received the UP-C intervention (see Figure 1). Of these, 43.8% were referred by school psychologists, 37.5% had self-registered on the project website and 18.8% were referred by mental health professionals at the hospital. Regarding follow-up, a total of 4 losses (13.3%) can be counted, with two dropouts (both children referred by the school), one family moved to another country and another family could not be contacted.

### 3.2. The Feasibility of the Data Collection Procedures

Although in the present study only measures of anxiety and depression and their interference were included, the complete baseline assessment protocol included other child variables (e.g., behavioral avoidance, temperament, and cognitive inflexibility) and parent variables (e.g., overprotection and neuroticism); therefore, it took approximately 45 min to complete the assessment. No changes were made to the content of the baseline assessment protocol since children and parents were able to understand and complete all measures.

The same battery of self-report questionnaires applied at baseline was planned to be applied in the subsequent assessments (midtreatment, posttreatment, and 3-month follow-up). However, parents and children reported feeling weariness with so many assessments and with the extensive assessment battery, and it was also possible to identify fatigue in children by clinical behavioral observation. Thus, we decided to shorten the assessment battery used at midtreatment and follow-up by excluding the measures of hypothesized predictor variables (e.g., temperament and neuroticism), which were only assessed at baseline. Therefore, the protocol at the midtreatment and follow-up took approximately 30 min to complete.

### 3.3. The Feasibility of the Implementation of the UP-C Intervention and Study Procedures

As initially planned, six intervention groups were formed for the present feasibility study (of the 34 eligible participants, 32 received the UP-C).

#### 3.3.1. Resources

As recommended in the original program [35], the intervention groups were planned to be led by two clinical psychologists trained in the UP-C and with experience in CBT intervention with children (one psychologist for the children’s group and another for the parents’ group). However, during the implementation of the first UP-C group intervention, we found that the groups were more easily and effectively managed if two psychologists were present in the children’s group (and not only one as initially planned), especially in the case of larger and/or more diverse groups (e.g., in terms of age or reading and writing abilities). For this reason, one intern of with a master’s degree in clinical psychology was added to the team to be included in the children’s group.

#### 3.3.2. Limitations Due to COVID-19

This study was conducted during the COVID-19 pandemic period, which brought some obstacles to the application of this face-to-face intervention. First, we decided to organize smaller groups than initially planned (4–7 individuals instead of 5–8 as recommended by the authors of the original program [35]). Moreover, three groups (group 2, 3, and 4) had their sessions temporarily interrupted due to the national lockdown that occurred from mid-January 2021 until the beginning of April 2021. Group 2 was interrupted at the 12th session, and Groups 3 and 4 were interrupted at the 5th session. In Group 2, as exposure sessions had already started (which occurred from session 10 to session 14), six video calls were made with the children and their parents to prescribe the exposure tasks to be completed at home and to monitor their progress. In April 2021, face-to-face sessions were resumed for a final exposure session and for completing Session 15 (celebration, relapse prevention, and future planning). As Groups 3 and 4 were interrupted at the 5th session, the intervention itself was suspended, and four video calls were made to maintain the therapeutic relationship. In April 2021, face-to-face sessions resumed, and a brief review of the contents of the first five sessions was made in the first post-lockdown session. Groups 1, 5, and 6 were not interrupted.

In all groups, including those that were not interrupted by the lockdown, some children missed one or more sessions because they were in prophylactic isolation. In these cases, families were allowed to participate in the group session through videoconference.

#### 3.3.3. Treatment Integrity

Therapist adherence to the UP-C was 95.4%, which means that the majority of the UP-C content was addressed by the therapists in all the sessions. There was 100% agreement between raters on UP-C sessions that were double coded. The sessions in which the therapists had more difficulty addressing all the content due to time constraints were Sessions 5 and 8 for parents and Session 15 for children and parents (see Table 3). Regarding Session 5 for parents, the unaddressed content was usually addressed at the beginning of Session 6. However, some unaddressed content of Sessions 8 and 15 were not addressed later, including the review of learned skills in Session 8 for parents (66.7% of the sessions), the explanation of the relapse concept in Session 15 for children (83.3% of the sessions), and the identification of five steps to continue working at home with parents in Session 15 for parents (50% of the sessions).

Moreover, although the UP-C sessions were planned to last 90 min, 54.5% of the sessions took longer (ranging from 100 to 120 min). Often, the parental sessions took longer than the children’s sessions. Children’s exposure sessions also usually took longer than 90 min.

#### 3.3.4. Learned Skills

According to the clinician report, the UP-C skills most easily learned by children were those related to psychoeducation about emotions (M = 1.41, SD = 0.71, range: 1–3) and emotional exposure techniques (M = 1.40, SD = 0.67, range: 1–3), followed by problem solving skills (M = 1.60, SD = 0.63, range: 1–4) and cognitive skills (M = 1.69, SD = 0.96, range: 1–4). The experiential and mindfulness skills were the ones where the children had more difficulty, although with a good level of learning (M = 2.23, SD = 1.07, range: 1–4).

### 3.4. Acceptability of the UP-C by Children and Parents

#### 3.4.1. Attendance and Dropout

Attendance ranged from 6 to 15 sessions (M = 14.09, SD = 1.51). Of the 32 children enrolled in the UP-C, 93.8% reached treatment completer status (defined as attending at least 70% sessions, i.e., at least 11 of the 15 sessions [53]). All absences were justified (due to illness, prophylactic isolation, and family events). Two participants dropped out of treatment (6.3%). At baseline, the total scores of RCADS and CALIS-C for the two dropout participants (RCADS = 37 and 59; CALIS-C = 32 and 21) did not differ markedly from the mean of the total sample (RCADS: M = 58.97, SD = 23.28; CALIS-C: M = 19.16, SD = 6.83). However, the total scores of CALIS-P for the two dropout participants (CALIS-P = 8 and 7) were markedly inferior to the mean of the total sample (M = 28.09, SD = 13.83). Both families were referred by the school psychologists, were from a rural area, and had a low socioeconomic level (the monthly income was lower than EUR 800, which was lower than the average monthly income of the sample of the present study). In addition, one child had learning difficulties, and both children dropped out at the early phase of the treatment (Sessions 6 and 7, respectively). Both cases belonged to groups that were interrupted (in Session 5) due to the COVID-19 pandemic.

#### 3.4.2. Session-to-Session Commitment

On average, 88.3% of the children’s homework and 80.4% of the parents’ homework were completed. Children and parents reported having talked about the UP-C between sessions in 94.3% of the treatment weeks (M = 94.32, DP = 0.13).

#### 3.4.3. Session-to-Session Motivation and Satisfaction

According to parents’ self-report, children and parents were, on average, very or extremely motivated to participate in the UP-C intervention (M = 4.61, SD = 0.57 and M = 4.84, SD = 0.29, respectively). Children’s and parents’ motivation seems to have gradually increased as the treatment progressed (see Figure 2).

Children and parents rated their satisfaction across all 15 sessions of the program. On average, children considered the sessions to be very fun or extremely fun (M = 4.75, SD = 0.33, range from 3.87 to 5), reported having understood almost everything or everything in the session (M = 4.31, SD = 0.45, range from 3.60 to 5), and considered the sessions to be “very useful/help me a lot” or “extremely useful/help me very much” (M = 4.64, SD = 0.48, range from 3.27 to 5). Parents, on average, considered the sessions very useful or extremely useful for their children (M = 4.71, SD = 0.42, range from 3.80 to 5) and very useful or extremely useful for themselves (M = 4.77, SD = 0.40, range from 3.92 to 5). Parents rated the psychologists’ knowledge and adequacy in the way they conveyed the sessions’ content as very adequate or extremely adequate (M = 4.88, SD = 0.31, range from 4 to 5). The children’s and parents’ weekly ratings are presented in Figure 2.

#### 3.4.4. End-of-Intervention Satisfaction

Parents and children reported a high degree of satisfaction on the End of Program questionnaire (M = 4.82, DP = 0.28, range: 4–5; and M = 4.74, DP = 0.36, range: 3.67–4.74, respectively). In a qualitative assessment of what children and parents liked the most and the least, we found that for children, the activities, games, awards, having been able to face their fears, the exposure sessions, the companionship with colleagues and the relationship with the psychologists were reported as what they liked the most. Most children answered the question of what did you like the least with “nothing” (81.3%), followed by having to complete the assessment questionnaires several times during the intervention (9.4%). Parents reported that what they liked most revolved around aspects such as seeing their children’s improvements throughout the intervention, learning parenting strategies, the UP-C content itself and the way psychologists conveyed it, the companionship between the children, the relationship of children and parents with the psychologists, and the sharing between parents. Most parents also answered the question what did you like least, with answers such as “nothing” (31.3%) or left it blank (40.6%), followed by the time/day of the intervention not being the most favorable (12.5%).

### 3.5. Effects on Treatment Outcomes

#### 3.5.1. Symptom Severity According to Top Problems (Child’s and Parent’s Report)

As presented in Figure 3, the severity ratings of children’s three top problems decreased throughout the intervention. At the beginning of treatment (in the first UP-C session), the mean values of the three top problems (children’s and parents’ reports) were greater than 6 (on a scale of 0 to 8). At the end of treatment (in the last week of the UP-C intervention), the mean values of the three top problems were lower than 3 (see Figure 3).

#### 3.5.2. Symptom’s Severity (Clinician Report)

At pretreatment, therapists rated 65.7% of children with an overall severity score between 5 and 7 (“5: among the most extremely ill of patients” (12.5%), “6: severely ill” (21.9%) or “7: markedly ill” (31.3%)). In addition, 25% of children were rated as “moderately ill”, and a minority were classified as “mildly ill” (9.4%). At posttreatment, the majority of children were rated with a severity score of 1 (“normal, not at all ill”; 40%) or 2 (“borderline mentally ill”; 33.3%). The remaining 26.7% of children received a severity score of 3 (“mildly ill”; 16.7%), 4 (“moderately ill”; 3.3%), or 5 (“markedly ill”; 6.7%).

To further examine treatment effects on severity ratings, a Wilcoxon test was performed, and the results indicated significant differences between the mean scores of the CGI-S at pretreatment and at posttreatment (Z = 4.85; *p* < 0.001), with significantly lower levels of severity at posttreatment (M = 2.03; DP = 1.16) than at pretreatment (M = 5.03; DP = 1.22).

#### 3.5.3. Treatment Improvement (Clinician Report)

Regarding the Improvement scale (CGI-I), at posttreatment, 50% of children were rated with a score of 1 (“very much improved”) and 33.3% with a score of 2 (“much improved”), which is considered to be indicative of treatment response. A minority (16.7%) presented a score of 3 (“minimally improved”).

#### 3.5.4. Children’s Anxiety, Depression and Symptom Interference in Children’s and Families’ Lives (Child’s and Parent’s Reports)

A series of repeated-measures ANOVAs were conducted to compare levels of anxiety and depression and of the interference of symptomatology in children’s and families’ lives at pretreatment, midtreatment, posttreatment, and 3-month follow-up. As presented in Table 4, the results revealed a significant main effect of time and a medium to large effect for the RCADS total and anxiety scores, as well as for both the CALIS-C and CALIS-P scores. A significant main effect of time and a small to medium effect was found for depression. As presented in Figure 4, the RCADS, CALIS-C, and CALIS-P scores decreased over time. As presented in Table 5, a statistically significant decrease from mid- to posttreatment was also found for all variables, with the exception of depression. Pairwise comparisons showed a statistically significant difference in all variables between pre- and posttreatment as well as between mid- and posttreatment, with significantly lower levels of children’s anxiety and depression and lower levels of the interference of the symptomatology in children’s and families’ lives at posttreatment (see Table 6).

Finally, as presented in Table 7, regarding the effects at the 3-month follow-up, statistically significant differences between pretreatment and follow-up and between midtreatment and follow-up were found for all variables. From posttreatment to follow-up, a slight increase in the mean values of anxiety and depression and a slight decrease in the mean values of the interference of symptomatology in children’s and families’ lives were observed, but these differences were not statistically significant (see Table 4 and Table 7).

## 4. Discussion

The UP-C is a transdiagnostic and emotion-focused CBT treatment for children’s EDs [35] that offers clinical and practical benefits compared with traditional CBT protocols, especially in cases of comorbidity. Although UP-C feasibility and efficacy studies are promising, the few existing studies are limited to the American population [36,37,38]. Hence, further research is needed to examine UP-C feasibility and efficacy, particularly in other cultures (such as in Europe). Therefore, the main goal of the present study was to examine the acceptability and feasibility of the UP-C among the Portuguese population and to analyze whether the UP-C study protocol required any modifications before proceeding to an RCT aimed at examining the efficacy of this intervention in the Portuguese population. Our results confirm the acceptability and feasibility of the UP-C in the Portuguese population and clearly support proceeding with the implementation of an RCT, providing important information on how to proceed.

Regarding the recruitment procedures, specifically the inclusion criteria, this feasibility study allowed us to verify that we can change the age criterion that was initially established, extending it to include 6-year-old children. Since we did not observe any difficulties in implementing the program with 6-year-olds, we decided to maintain this criterion in the RCT. Furthermore, in the current study, it was possible to realize that to obtain the number of participants needed for an RCT study, it may not be feasible to collect the sample only from public mental health institutions, considering that a request for collaboration was made to several central hospitals and we were only given approval by one of them. One of the explanations can be due to the pandemic situation caused by COVID-19 of which the end is uncertain. Thus, in the RCT we will maintain the possibility of the self-registration of parents on the study website, as well as the procedure of screening children who were self-registered on website or referred by the psychologists of the collaborating schools. However, it is important to note that although these procedures resulted in a higher number of potential participants, a large proportion of cases were subsequently excluded for not meeting the eligibility criteria. We hypothesize that for children referred by school psychologists this may have happened because, in Portugal, most school psychologists are trained in educational psychology and not in clinical psychology; moreover, it may also be due to the fact that school psychologists may receive requests from teachers or other educational agents who may not be aware of the differential diagnosis between the different disorders. In contrast, the cases referred by mental health professionals, although few, were well-screened. In an attempt to minimize the number of misdirected cases, some measures were taken. For instance, a mandatory field was created on the study website for telephone contact information so that the research team could contact parents and thus provide information about the research project and assess whether the child was a suitable candidate to receive the UP-C intervention. A similar screening phone call was made for all parents of children referred by the school psychologists. This procedure seems to be effective in ensuring that the applicants are eligible for the study and reducing the loss of participants due to the impossibility of being contacted. Therefore, this procedure will be implemented in a future RCT.

Regarding the assessment protocol, since participants often reported tiredness in completing the questionnaires, the assessment protocol was shortened in the midtreatment, posttreatment, and follow-up assessments. Although many participants also mentioned that the number of evaluations was slightly excessive, we decided to maintain the same number of evaluations in the RCT so that we could examine change processes.

With regard to the implementation of the UP-C, the initially planned procedures proved to be feasible. As planned, the intervention groups started between less than one week and four weeks after the baseline assessment. In addition, it was possible to implement the number of group interventions and obtain the number of participants that were initially planned within the original timeframe (even with the limitations associated with the COVID-19 pandemic). Throughout this study, we found that a larger team of therapists was needed to more effectively implement the UP-C groups (ideally, three therapists per intervention group: two in the children’s group and one in the parents’ group); hence, this should be provided in a future RCT. In addition, the fact that psychologists were able to apply almost all the content of the UP-C in all sessions (high therapist adherence) demonstrates the feasibility of the UP-C implementation procedures. However, almost all sessions took longer than the scheduled 90 min, especially the parents’ sessions. Perhaps in a future RCT study, a two-hour session can be considered, and/or some group dynamics can be introduced in the children’s group while the therapist of the parents’ group wraps up the parents’ session. Regarding the therapists’ assessment of the UP-C skills learned by the children, we found that all skills were well-learned by the children overall. The skills that were assessed as the most difficult to learn were mindfulness skills. This may be due not only to the characteristics of the competencies themselves but also to the relatively little time spent in the program for their learning. A possible strategy to overcome this limitation can be to extend sessions 8 from 90 min to 2 h so that these skills can be more easily learned.

The UP-C was found to be an acceptable intervention in our sample. Despite being a long intervention (15 sessions), with a weekly frequency and strong parental involvement (parents participated in almost all sessions), dropout rates were very low (6.3%), and attendance rates were very high (ranging from 6 to 15 sessions; M = 14.09, SD = 1.51), with 93.8% of the participants reaching the status of treatment completers. Dropout rates were lower than what is typically reported in feasibility studies of cognitive-behavioral interventions for anxious children, in which dropout rates may range between 7% and 27% [54]. Likewise, adherence rates were higher than those obtained in previous studies, namely, in a previous UP-C open trial in which attendance ranged from 3 to 15 sessions and only 73% of children reached treatment completer status [36].

In addition, we also found a high rate of completion of weekly home learning assignments (88.3% for children and 80.4% for parents), which also demonstrates a strong commitment to the treatment and the acceptability of the intervention. Furthermore, parents and children reported having talked with each other about the UP-C between almost every session (94.3%), which also demonstrates the interest of families in the intervention.

Parents’ and children’s levels of satisfaction with different aspects of the intervention were shown to be high. For instance, at the end of each session, parents and children were asked to rate the session in terms of aspects such as fun, understanding, and usefulness of session content, as well as to report their level of motivation to participate in this intervention. According to the parents’ self-reports throughout the sessions, children and parents were, on average, very or extremely motivated to participate in the intervention. Additionally, on average, children rated each session as very or extremely useful and fun and reported having understood all or almost all the content of the session. Similarly, parents rated the sessions as very or extremely useful for themselves and their children and rated the psychologists’ level of knowledge about the session content as very or extremely high. Moreover, at the end of the intervention, parents and children reported higher levels of overall satisfaction with the UP-C intervention.

The evolution of the top three problems identified by the children and their parents at the beginning of treatment as well as the levels of children’s anxiety and depression and their severity and interference were also assessed throughout the treatment.

First, the severity of the top three problems (according to both children’s and parents’ reports) markedly decreased over time (e.g., on a severity scale of 0–8, in the first session, ratings were on average higher than 6, and in the last session, ratings were on average lower than 3).

Second, symptom severity according to the therapists’ reports (in CGI) also significantly decreased from pre- to posttreatment, with the large majority of children receiving severity ratings of 1 (40%) or 2 (33.3%) posttreatment, which means that most children were classified as “normal, not at all ill” or “borderline mentally ill”, an indication of diagnostic remission [16,47]. Additionally, the majority of children received scores of 1 (“very much improved”; 50%) and 2 (“much improved”; 33.3%) on the CGI Improvement scale, which is considered to be indicative of treatment response [10,37].

Third, we found that anxiety and depression levels (child’s report), as well as their interference in the child’s and family’s life (child’s and parent’s report), significantly decreased over time, which is consistent with previous UP-C studies [36,37]. These improvements seem to be maintained after 3 months of follow-up. It is interesting to note that significant improvements were also found for depression, even though depression was not the main diagnosis of most children (only one child had depression as the main diagnosis).

We also found that from pre- to midtreatment (8th session), there was already a statistically significant decrease in anxiety and in its interference in the child’s life, although this was not found for depression; this may indicate that halfway through the treatment, some improvements in some variables can already be achieved. The first part of the UP-C encompasses psychoeducation, cognitive restructuring, problem solving and mindfulness techniques, which are techniques that have been shown to be effective in reducing children’s anxiety and depressive symptoms [55]. However, for significant reductions in depression levels to be found, the entire UP-C intervention seems to be needed. Furthermore, the levels of anxiety and of the interference of symptoms in the child’s and family’s life continued to decrease in a statistically significant way over time, which underlines the need to provide the complete UP-C intervention. The second part of the intervention is essentially focused on the application of the previously learned strategies, with a strong exposure component. From our clinical perspective, this part of treatment is the one in which children and parents express more significant and visible changes in the children’s behavior. Moreover, the exposure sessions were reported by children at the End of Program questionnaire as one of the components of the treatment that they liked the most. Congruently, at the session-to-session evaluation, the exposure sessions were the type of sessions best evaluated by the children and parents (e.g., usefulness, satisfaction, motivation, and fun).

The results of the present study attest the feasibility and acceptability of the UP-C for the Portuguese population, contributing to broaden the UP-C studies outside the USA and contributing for the availability of evidence-based treatment for children with ED in Portugal, being the first transdiagnostic, emotion-focused, parent-child and group intervention. Besides these contributions, there are several limitations that must be acknowledge. Since this is a feasibility study, the present study only allows the assessment of the acceptability and viability of the UP-C. Therefore, due to the sample size and the single-armed design (without a control group and without randomization), this kind of study does not allow the assessment of the efficacy of the UP-C. Thus, the results regarding the potential effect of the UP-C in reducing the levels of child’s anxiety and depression and its interference and severity are limited, absolutely needing a future RCT to analyze treatment effects. Particularly, without a control group it is not possible to analyze whether the reduction in children’s anxiety and depression levels and its interference and severity are a result of the treatment (treatment effect) or a result over time. In addition, the interference caused by the COVID-19 pandemic further compromised the conclusions about the possible efficacy of the UP-C.

A future RCT would allow the assessment of the efficacy of the UP-C in decreasing ED symptoms. Moreover, it would be interesting, in future studies, to examine the comparison of the UP-C with other types of intervention, to examine the changes in transdiagnostic variables (e.g., avoidance, emotion regulation) that are expected to underlie changes in emotional symptoms, as well as the changes in parental emotional behaviors targeted in this treatment (e.g., overprotection, criticism).

Additionally, other parental individual variables may be analyzed in more detail in a future RCT as predictors (e.g., children’s and parent’s age; parent’s psychopathology, socioeconomical level, etc.). For example, in the present study half of the parents had a diagnosed psychopathology (43.8%). Several authors argue that children of parents who suffer from psychological disorders or who exhibit extreme behavior in the presence of the child are vulnerable and at risk of developing mental health problems [56]. Such problems are caused, at least in part, by the important influence of modeling. Thus, it might be important in future studies to analyze the predictive role of these kind of variables in the child’s psychopathology and even in treatment adherence itself.

## 5. Conclusions

The results of the present study support the feasibility and acceptability of the UP-C in the Portuguese population and clearly support progress toward conducting an RCT to examine its efficacy. Moreover, this feasibility study provided important information on several practical issues that should be considered in a future study (e.g., sample collection procedures, number of professionals needed, age range to be included). In addition to examining the UP-C’s efficacy in decreasing ED symptoms, it would also be interesting if future studies examine changes in transdiagnostic variables underlying ED, changes in parental emotional behaviors targeted in this treatment and the predictive role of individual variables.

## Figures and Tables

**Figure 1 ijerph-19-01782-f001:**
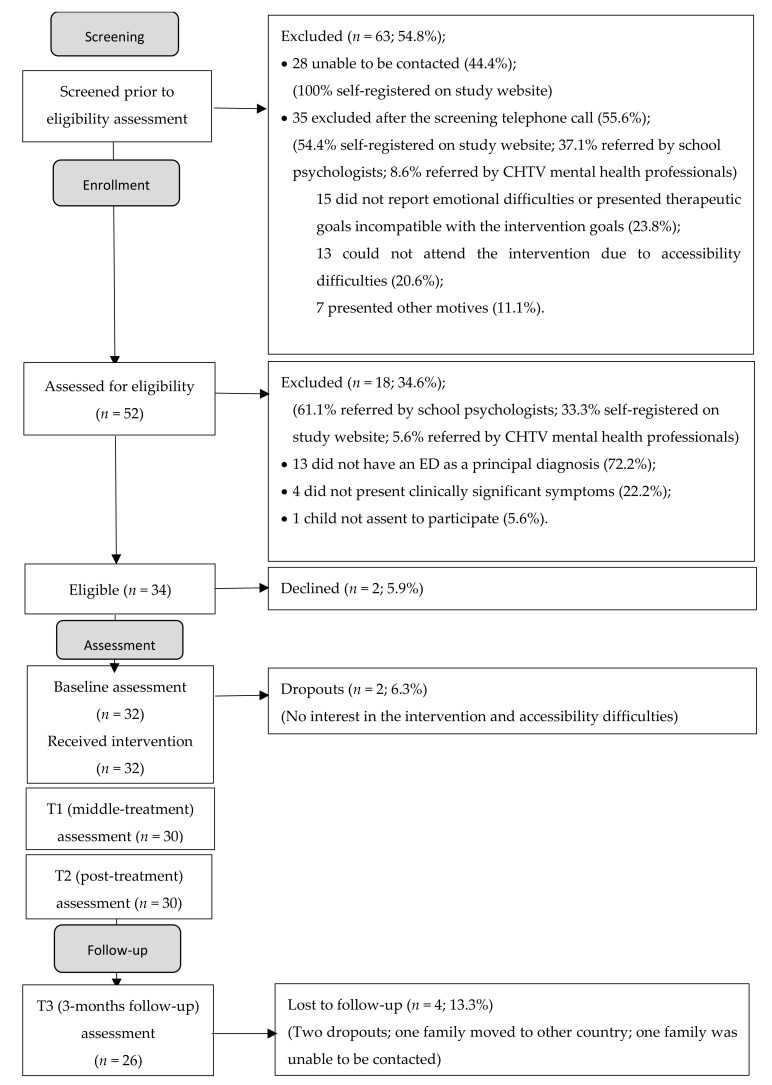
Participants Flow Diagram. Note: flowchart adapted by the Consolidated Standard of Reporting Studies Group (CONSORT; Eldridge, Chan, Campbell, Bond, Hopewell, Thabane, et al. CONSORT 2010).

**Figure 2 ijerph-19-01782-f002:**
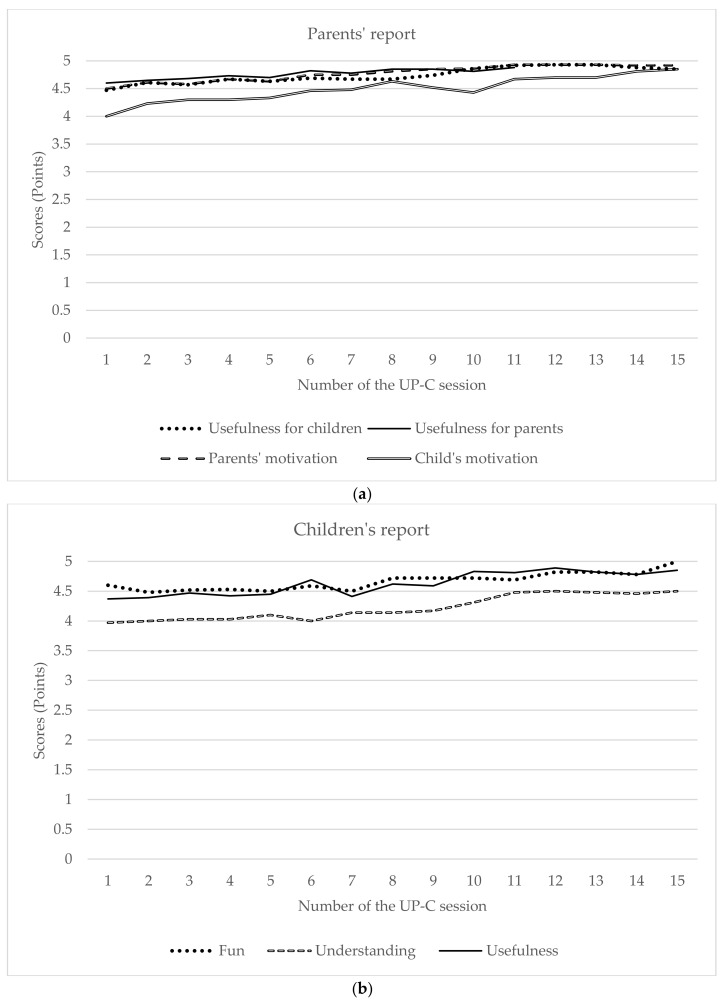
Session-to-session motivation and satisfaction (parents’ (**a**) and children’s report (**b**)).

**Figure 3 ijerph-19-01782-f003:**
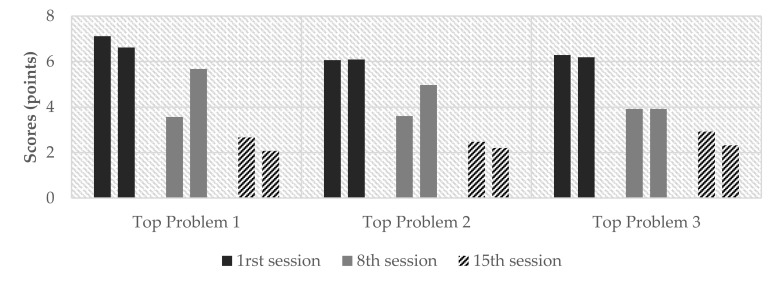
Children’s and parents’ reports on the severity of the three top problems throughout the UP-C intervention. Note. C: children’s report; P: Parents’ report.

**Figure 4 ijerph-19-01782-f004:**
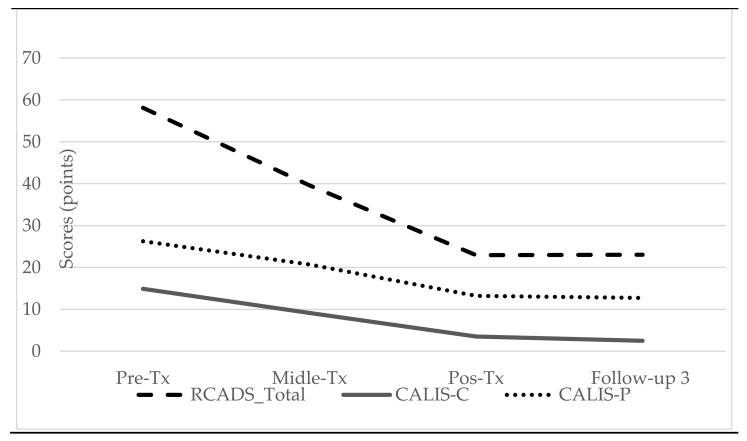
The evolution of children’s symptomatology (RCADS total score) and of the interference of the symptomatology in child’s and family life, according to the child’s report (CALIS-C) and parent’s report (CALIS-P).

**Table 1 ijerph-19-01782-t001:** Children’s principal diagnoses at pretreatment.

Principal Diagnosis	Comorbidity with Another ED	%
Anxiety Disorder	Another Anxiety Disorder	35.7%
	Anxiety Related Disorder	3.6%
	Depression	21.4%
Anxiety-Related Disorder	Anxiety Disorder	66.7%
	Depression	33.3%
Depression	Anxiety Disorder	100%

**Table 2 ijerph-19-01782-t002:** Children’s comorbid diagnoses at pretreatment.

Principal Diagnosis	(%)
Anxiety Disorder	87.5%
Specific Phobia	25%
Separation Anxiety Disorder	18.8%
Generalized Anxiety Disorder	18.8%
Social Phobia/Performance Anxiety	12.6%
Selective Mutism	6.3%
Anxiety Disorder not Otherwise Specified	3.1%
Panic Disorder	3.1%
Anxiety-Related Disorder	9.4%
Obsessive-Compulsive Disorder	6.3%
Illness Anxiety Disorder	3.1%
Depression	3.1%

**Table 3 ijerph-19-01782-t003:** UP-C treatment contents, CLUES.

	Children’s Contents	Parents’ Contents
Consider How I Feel (Sessions 1–4)	Psychoeducation about emotions; behavioral activation; body scanning; interoceptive exposure.	Psychoeducation about emotions and about emotional parenting behaviors vs. opposite parenting behaviors; positive reinforcement vs. criticism; empathy.
Look at My Thoughts (Session 5)	Psychoeducation; thought identification; identification of thinking traps.	Reinforcement and punishment; consistent discipline and praise vs. inconsistency.
Use Detective Thinking (Sessions 6–7)	Cognitive flexibility; problem solving.	Promotion of healthy independence vs. overprotection.
Experience My Fears and Feelings (Sessions 8–14)	Present moment awareness; non-judgmental awareness, situational emotion exposures.	Exposure vs. avoidance; parental healthy emotional modelling vs. excessive modelling of intense emotions and avoidance; support children’s exposure vs. modelling of avoidance.
Stay Healthy and Happy (Session 15)	Competency review; create plan for the future; celebration of the process and gains.	Competency review generalization of gains and relapse prevention; create a plan for the future; lapse vs. relapse.

Note: In addition to the specific contents learned by the parents (opposite parenting behaviors), parents also learned the same strategies as children.

**Table 4 ijerph-19-01782-t004:** Descriptive Statistics, mean differences between time-points and effect size estimates.

Measure	Pre-Tx M (SD)	Middle-Tx M (SD)	Post-Tx M (SD)	Follow-Up M (SD)	F (1, 3)	n_p_^2^
RCADS Total	58.11 (4.22)	39.59 (3.83)	22.93 (3.03)	23.04 (3.42)	29.50 **	0.79
RCADS Anxiety	49.93 (3.59)	33.07 (3.35)	18.74 (2.57)	18.78 (2.72)	35.45 **	0.82
RCADS Depression	8.19 (0.83)	6.52 (0.74)	4.19 (0.64)	4.26 (0.84)	6.86 *	0.46
CALIS-C	14.89 (1.09)	9.15 (1.06)	3.49 (0.71)	2.48 (0.57)	52.95 **	0.87
CALIS-P	26.25 (2.46)	20.67 (2.37)	13.21 (2.08)	12.71 (2.06)	14.03 **	0.67

**p* < 0.05; ** *p* < 0.01. Note: RCADS Total: scale total score; RCADS Anxiety: anxiety total score; RCADS Depression: depression total score.

**Table 5 ijerph-19-01782-t005:** Effects at middle-treatment.

Variable	Pre-Tx–Middle-Tx Mean Difference [95% CI]
RCADS Total	18.52 ** [8.91; 28.13]
RCADS Anxiety	16.85 ** [8.34; 25.37]
RCADS Depression	1.67 [−0.26; 3.60]
CALIS-C	5.74 ** [3.36; 8.13]
CALIS-P	5.58 * [1.08; 10.09]

**p* < 0.05; ** *p* < 0.01.

**Table 6 ijerph-19-01782-t006:** Effects at posttreatment.

Variable	Pre-Tx–Post-Tx Mean Difference [95% CI]	Middle Tx–Post-Tx Mean Difference [95% CI]
RCADS Total	35.19 ** [24.70; 45.67]	2.81 ** [8.66; 24.68]
RCADS Anxiety	31.19 ** [22.64; 39.73]	14.33 ** [7.65; 21.02]
RCADS Depression	4.00 * [1.52; 6.48]	2.33 * [0.26; 4.41]
CALIS-C	11.41 ** [8.63; 14.18]	5.67 ** [3.01; 8.33]
CALIS-P	13.04 ** [7.46; 18.62]	7.46 ** [3.05; 11.87]

**p* < 0.05; ** *p* < 0.01.

**Table 7 ijerph-19-01782-t007:** Effects at 3-months follow-up.

Variable	Pre-Tx–FP-Tx Mean Difference [95% CI]	Middle-Tx–FP-Tx Mean Difference [95% CI]	Post-Tx–FP-Tx Mean Difference [95% CI]
RCADS Total	35.07 ** [23.53; 46.62]	3.68 ** [6.06; 27.05]	0.11 [−7.16; 6.94]
RCADS Anxiety	31.15 ** [21.54; 40.75]	14.30 ** [5.23; 23.36]	−0.37 [−6.11; 6.04]
RCADS Depression	3.93 ** [1.33; 6.52]	2.26 * [0.18; 4.34]	−0.07 [−1.59; 1.45]
CALIS-C	12.41 ** [9.54; 15.27]	6.67 ** [3.48; 9.86]	1.00 [−1.06: 3.06]
CALIS-P	13.54 ** [6.91; 20.17]	7.96 * [2.50; 13.42]	0.50 [−2.69; 3.69]

**p* < 0.05; ** *p* < 0.01.

## Data Availability

Not applicable.

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
