# Peer review of "The Unified Protocol for Transdiagnostic Treatment of Emotional Disorders in Children (UP-C) in Portugal: Feasibility Study Results"

_ijerph, 2022, doi:10.3390/ijerph19031782_

Round 1

Reviewer 1 Report

The manuscript provides data concerning the feasibility of a trans-diagnostic psychological intervention focusing on emotional disorders in children and adolescents.

This is an interesting manuscript, but I have some points to raise to the authors.

1) Emotional Disorders are not a recognized entity described as such in any international classification system. The authors should therefore describe in detail what they consider to be or not an Emotional Disorder.

2) It is important to avoid unreferenced affirmations. To make an example: I think that CBT interventions are probably highly prescribed, but not everywhere in the world and not the same kind of CBT interventions.

3) The authors state that verbal assent was obtained from children involved in the study. Why not a written assent?

4) Was an a-priori power calculation obtained? How or why not?

5) I can't understand how where the 6 groups obtained. According to the authors, in fact, there were: "one at the hospital, two at the schools, and one at the Faculty of Psychology and Education Sciences of the University of Coimbra". This makes either 4 groups or 8 groups (if two per each of the 3 schools).

6) Adding "non-emotional disorders" as acceptable comorbidities might have increased generalizability, but I wonder: 1) how the authors managed to have group interventions with children with so many highly relevant clinical differences and 2) if this has not moved the intervention to a "one-size-fits-all" level.

7) Please use DSM-5 and not DSM-V.

8) If pervasive developmental disorders were an exclusion criterion, how could you accept children with Autism Spectrum Disorder as comorbidity?

9) "18.8% presented comorbidity with behavioral disorders": please clarify these comorbidities. Was the impact of comorbidities studied? How or why not?

10) I think that the authors assessed variables distribution, but this is not described in the methods.

11) Where differences between referred patients and spontaneous accesses assessed? How or why not?

12) About half of the parents had a diagnosed psychopathology. More informations are needed on this and I wonder if this was used in the statistical analysis and how.

13) The number of drop-outs is unclear for self-accessing families.

14) Percentages could be given with one decimal (and please avoid things like "50.00%").

15) A number of changes where done to the original protocol due to the COVID pandemia. This is reasonable and understandable, but I wonder if these differences where considered in the statistical analysis.

16) If the therapist had an adherence as high as 95.4%, I wonder why (in the author's words): "although the UP-C sessions were planned to last 90 minutes, 54.5% of the sessions took longer (ranging from 100 to 120 minutes). Often, the parental sessions took longer than the children’s sessions. Children’s exposure sessions also usually took longer than 90 minutes".

17) Where the 3 top problems the same for all childrens? Otherwise, please explain why they were grouped together.

18) No limitations are given.

Author Response

First of all, we would like to thank the reviewers and editor for the opportunity to review this manuscript thank you for your kind comments and suggestions that were very relevant to improving the manuscript.

We believe that the present study is relevant since this is the first study studying the Unified Protocol for Children outside the USA. This is a transdiagnostic intervention for the treatment of children’s emotional disorders that we believe contributes largely to evidence-based CBT psychological interventions.

All suggestions given by the reviewers and editor were taken into account. Please find above in this letter our answers explaining how we addressed to these suggestions. Changes were made to the word document (marked in yellow) and comments will be answered below.

Please find attached the revised manuscript which we believe that has improved considerably with these changes. Thus, we hope that the manuscript can be accepted for publication!

COMMENT 1: Emotional Disorders are not a recognized entity described as such in any international classification system. The authors should therefore describe in detail what they consider to be or not an Emotional Disorder.

ANSWER: As we wrote at the manuscript “EDs include anxiety disorders (e.g., separation anxiety disorder), anxiety-related disorders (e.g., obsessive-compulsive disorder) and mood disorders (e.g., major depressive disorder) [4,5]”. However, we clarified this issue at the manuscript with a more detailed explanation.

COMMENT 2: The authors state that verbal assent was obtained from children involved in the study. Why not a written assent?

ANSWER: Children are defined as persons who have not attained the legal age for consent to treatments or procedures involved in clinical investigations under the applicable law of the jurisdiction in which the clinical investigation will be conducted (see. https://www.fda.gov/science-research/clinical-trials-and-human-subject-protection/additional-protections-children). Because children are unable, due to age, to give consent themselves, permission is provided by a parent or guardian on their behalf.

COMMENT 3: Was an a-priori power calculation obtained? How or why not?

ANSWER: The Consolidated Standards of Reporting Trials Group and bodies such as The National Institute for Health Research and The National Research Ethics Service state that not all studies necessarily need a power-based sample size calculation but they do all need a sample size justification. Therefore, since the purpose of the pilot is not to assess the efficacy of the intervention, then the sample size provided by the conventional calculations may be higher than necessary (Lancaster, Dood & Williamson, 2004). To estimate the sample size of this study we follow the rules suggested by Browne (1995), that recommended a minimum of 30 participants in a pilot trial. Since after the implementation of this feasibility study we intend to conduct a randomized controlled trial with the UP-C, we chose to reduce the sample to the minimum number of participants recommended and stopped data collection after reaching about 30 participants.

COMMENT 4: I can't understand how where the 6 groups obtained. According to the authors, in fact, there were: "one at the hospital, two at the schools, and one at the Faculty of Psychology and Education Sciences of the University of Coimbra". This makes either 4 groups or 8 groups (if two per each of the 3 schools).

ANSWER: Thank you in advance for the correction, there was indeed a mistake at the manuscript. The correction has already been made in the current manuscript.

COMMENT 5: Adding "non-emotional disorders" as acceptable comorbidities might have increased generalizability, but I wonder: 1) how the authors managed to have group interventions with children with so many highly relevant clinical differences and 2) if this has not moved the intervention to a "one-size-fits-all" level.

ANSWER: Thank you for your comment! First, we decided to include children with comorbidities because it had also been done in other UP-C studies, such as in the RCT of Kennedy et. al., 2019 that include anxious/depressed children with a variety of comorbidities (e.g., attention-deficit/hyperactivity disorder, oppositional defiant disorder, tic disorders) to increase sample representativeness and the generalizability of the results to real-world clinical settings (e.g., community/hospital settings). It should also be noted that the comorbidity between Emotional Disorders and, for example, Behavioral Disorders is high (e.g. Agati et al., 2019). In other words, choosing to include only children with Emotional Disorders in this study would be cleaner from a research point of view, but would not correspond to what the real needs are. Besides, in the case of our specific sample, situations of comorbidity are a minority: "18.8% presented comorbidity with behavioral disorders, and 3.1% presented comorbidity with another disorder (e.g., autism spectrum disorder)". Regarding the concern that including comorbid cases may move the intervention to a "one-size-fits-all" level, we believe that being a transdiagnostic intervention, focused on emotion and not on specific diagnoses, it is natural that there may even be improvements in symptoms other than just the emotional ones. However, the purpose of including comorbid cases is not necessarily to improve comorbid symptoms, but rather to show that, regardless of comorbidities with non-emotional disorders, this intervention works to treat emotional disorders. This type of study does not allow us to draw such conclusions (only an RCT would allow it) but that is the purpose of this intervention.

COMMENT 6: Please use DSM-5 and not DSM-V.

ANSWER: The suggested correction was made in the manuscript.

COMMENT 7: If pervasive developmental disorders were an exclusion criterion, how could you accept children with autism spectrum disorder as comorbidity?

ANSWER: Thank you for your relevant comment which we absolutely understand. Indeed, this was a topic discussed by our team when defining the eligibility criteria for this study. As a result of this discussion, we defined that Intense developmental disorders would be excluded, as well as cognitive deficit, whenever these would compromise the understanding of the UP-C program or in the case where emotional disorders are not the primary difficulty. In the specific case of Autism Spectrum Disorders, if the child has marked cognitive difficulties, marked impulsivity, and the focus of the family's concern is these difficulties directly associated with the disorder, it does not make sense to integrate into the program. However, if the child has an Autism Spectrum Disorder where the cognitive level is normative or higher, the child is globally adjusted, and the main difficulties are related to fear/anxiety, then the child is considered to benefit from the intervention. Of course, it would be easier to exclude all children with this type of diagnosis. However, thinking about what the objective of this intervention is (to work transdiagnostically with emotional difficulties regardless of diagnosis) it made a lot of sense that we could include children with Autism Spectrum Disorder who had these difficulties. And, indeed, our experience has shown us that these children have been very successful in the intervention and have benefited greatly from it.

COMMENT 8: "18.8% presented comorbidity with behavioral disorders": please clarify these comorbidities. Was the impact of comorbidities studied? How or why not?

ANSWER: As a response to your comment, in the manuscript, behaviour disorders has been replaced by Attention-Deficit/Hyperactivity Disorder (ADHD) which was actually the specific diagnosis presented by these children in this study.

Initially, the impact of comorbidities was explored (and it was found that the results were independent of the presence or absence of comorbidities). However, we chose not to include these results in this study, since this is not the purpose of a feasibility study. Thus, we left this type of analysis for the future RCT that is already in course.

COMMENT 9: I think that the authors assessed variables distribution, but this is not described in the methods.

ANSWER: Thank you! We now mention this aspect in the methods section.

COMMENT 10: Where differences between referred patients and spontaneous accesses assessed? How or why not?

ANSWER: The differences between referred patients and spontaneous accesses were only analyzed qualitatively, through the flow chart, and are explained in the discussion. We thought that a more detailed and quantitative analysis may not be pertinent or accurate given the small number of patients in this study.

COMMENT 11: About half of the parents had a diagnosed psychopathology. More informations are needed on this and I wonder if this was used in the statistical analysis and how.

COMMENT 12: The number of drop-outs is unclear for self-accessing families.

ANSWER: The two dropouts were referred by the school, so during treatment there were no dropouts of self-referred children, as explained in the manuscript. In any case, we added more explicit information regarding the two cases that dropped out before the start of treatment and the two cases that dropped out at follow-up.

COMMENT 13: Percentages could be given with one decimal (and please avoid things like "50.00%").

ANSWER: Thanks for the comment, the percentages have been corrected in the manuscript.

COMMENT 14: A number of changes where done to the original protocol due to the COVID pandemic. This is reasonable and understandable, but I wonder if these differences where considered in the statistical analysis.

ANSWER: In fact, it would be interesting to assess whether there were differences in the groups where changes were implemented and the groups where they were not, however we do not think we have a sufficient number of groups for such a comparison (3 groups where changes were made and 3 groups where no changes were made). Furthermore, the changes that were made were not exactly the same in all 3 groups. Because the objective of this study was essentially to evaluate feasibility and not efficacy, we considered this not to be a problem, although if we were talking about an RCT, this variable would clearly need to be controlled for.

COMMENT 15: If the therapist had an adherence as high as 95.4%, I wonder why (in the author's words): "although the UP-C sessions were planned to last 90 minutes, 54.5% of the sessions took longer (ranging from 100 to 120 minutes). Often, the parental sessions took longer than the children’s sessions. Children’s exposure sessions also usually took longer than 90 minutes".

ANSWER: When we refer to 95.4% adherence, we're referring to the adherence to the contents and not to the time the program took to be applied. Regarding the time of the sessions, we observed that for example, the ones that took the longest with the children were the exposure sessions, which don't even have structured contents, they are more individually prepared sessions and according to the specific group we have. In our opinion we consider that these are two different concepts.

COMMENT 16: Where the 3 top problems the same for all childrens? Otherwise, please explain why they were grouped together.

ANSWER:  The 3 top problems are defined by each family, translating their specific difficulties or their therapeutic goals. So, the top 3 problems are different from child to child. We clarified this at the manuscript, thank you. Children were not grouped by their top problems. Children were grouped according to the time they enrolled and their city of residence. Since the UP-C is a transdiagnostic intervention, groups can and should be diversified in terms of what their emotional difficulties are. If we grouped the children by the type of difficulty, we would be biasing the study.

COMMENT 18: No limitations are given.

ANSWER: Thank you so much! We added the limitations at the discussion point.

Reviewer 2 Report

This report reviewed the results of a single-arm feasibility study testing UP-C’s feasibility and acceptability in a sample of children with emotional disorders in Portugal. The paper is well written but far too long given the study was only a feasibility report.
1.    The sections 3.1 – Feasibility of the recruitment procedures; section 3.2 Participant Flow and 3.3.2 Limitations due to Covid-19 were much too detailed and a higher level summary of the issues would be more helpful to the readership.
2.    Tables 5, 6 and 7 were not necessary to include and the authors should focus more on the effect size results.
3.    The authors did not explain why the school psychologists struggled to refer appropriate participants and why it was not feasible to collect participants from public mental health institutions? Were these difficulties unique to implementing the study in Portugal?
4.    On figures 3 and 4, the authors have not indicated the units of measurement on the Y axis. 

Author Response

First of all, we would like to thank the reviewers and editor for the opportunity to review this manuscript thank you for your kind comments and suggestions that were very relevant to improving the manuscript.

We believe that the present study is relevant since this is the first study studying the Unified Protocol for Children outside the USA. This is a transdiagnostic intervention for the treatment of children’s emotional disorders that we believe contributes largely to evidence-based CBT psychological interventions.

All suggestions given by the reviewers and editor were taken into account. Please find above in this letter our answers explaining how we addressed to these suggestions. Changes were made to the word document (marked in yellow) and comments will be answered below.

Please find attached the revised manuscript which we believe that has improved considerably with these changes. Thus, we hope that the manuscript can be accepted for publication!

COMMENT 1: The sections 3.1 – Feasibility of the recruitment procedures;  section 3.2 Participant Flow and 3.3.2 Limitations due to Covid-19 were much too detailed and a higher level summary of the issues would be more helpful to the readership.

ANSWER: We understand your suggestion! We've tried to summarize the information in section 3.1 and 3.3.2., we hope it's better now! Section 3.2 was not able to be summarized as we would like as it seems to us to be relevant information for a feasibility study and, at the same time, to be able to respond to the suggestions of reviewer 1. Thank you!

COMMENT 2: Tables 5, 6 and 7 were not necessary to include and the authors should focus more on the effect size results.

ANSWER: Thank you for your suggestion that deserved our reflection! We agree with a greater focus on effect size results, however, we would like to keep tables 5, 6 and 7, since, in the future, we would like to can compare these results with the results that we will achieve in the RCT study (which is currently underway). We hope you understand, thank you!

COMMENT3:   The authors did not explain why the school psychologists struggled to refer appropriate participants and why it was not feasible to collect participants from public mental health institutions? Were these difficulties unique to implementing the study in Portugal?

ANSWER: Regarding the difficulty that was felt in the referral of children by school psychologists, we added in the discussion a brief explanation for the hypothesis we raised that this happened. As for the fact that we think it is not possible to sample only through public mental health institutions, we have made a small change in our statement and have also provided a brief justification in the article.

COMMENT 4:   On figures 3 and 4, the authors have not indicated the units of measurement on the Y axis. 

ANSWER: Thank you for the correction, we have made the change in the manuscript.

Round 2

Reviewer 1 Report

I think the authors did a very good work in answering to my previous concerns. I still think that a written assent (not consent, which is given by the parents) from the children (when possible) could have been a better choice, but I also believe that the present version of the manuscript deserves to be published.

Reviewer 2 Report

The revised article has address most of the reviewer's concerns.